# Technological Research of a Clean Energy Router Based on Advanced Adiabatic Compressed Air Energy Storage System

**DOI:** 10.3390/e22121440

**Published:** 2020-12-20

**Authors:** Chenyixuan Ni, Xiaodai Xue, Shengwei Mei, Xiao-Ping Zhang, Xiaotao Chen

**Affiliations:** 1Department of Electronic, Electrical and Systems Engineering, The University of Birmingham, Birmingham B15 2TT, UK; CXN603@student.bham.ac.uk (C.N.); X.P.Zhang@bham.ac.uk (X.-P.Z.); 2China State Key Laboratory of Power System and Generation Equipment, Department of Electrical Engineering, Tsinghua University, Beijing 100084, China; xuexiaodai@mail.tsinghua.edu.cn; 3Qinghai Key Lab of Efficient Utilization of Clean Energy (New Energy Photovoltaic Industry Research Center), Qinghai University, Xining 810016, China; meishengwei@tsinghua.edu.cn

**Keywords:** energy Internet, advanced adiabatic compressed air energy storage, clean energy router, comprehensive energy utilization

## Abstract

As a fundamental infrastructure of energy supply for future society, energy Internet (EI) can achieve clean energy generation, conversion, storage and consumption in a more economic and safer way. This paper demonstrates the technology principle of advanced adiabatic compressed air energy storage system (AA-CAES), as well as analysis of the technical characteristics of AA-CAES. Furthermore, we propose an overall architectural scheme of a clean energy router (CER) based on AA-CAES. The storage and mutual conversion mechanism of wind and solar power, heating, and other clean energy were designed to provide a key technological solution for the coordination and comprehensive utilization of various clean energies for the EI. Therefore, the design of the CER scheme and its efficiency were analyzed based on a thermodynamic simulation model of AA-CAES. Meanwhile, we explored the energy conversion mechanism of the CER and improved its overall efficiency. The CER based on AA-CAES proposed in this paper can provide a reference for efficient comprehensive energy utilization (CEU) (93.6%) in regions with abundant wind and solar energy sources.

## 1. Introduction

With the continuous growth of global energy production and consumption as well as the exhaustion of traditional fossil fuels, posing a great threat to the survival and development of mankind, the development of renewable energy (RE) has become a common challenge facing the world. However, RE has the characteristics of volatility and intermittence under the influence of natural conditions, which is not conducive to its effective control and limits the popularization and application of RE. To overcome the above problems, the research and development of new technologies have attracted great attention from academic and industrial circles around the world, especially concerning RE such as solar and wind energy, as well as the replacement of traditional fossil energy by clean energy to obtain a safe, economical, and clean energy supply system [1,2,3].

However, with the increasing scale of RE and the development of distributed generation (DG), the intermittent and volatile nature of power generation has become increasingly prominent. Therefore, there is an urgent need for effective technical solutions to solve the problem of network peak frequency modulation caused by the large-scale interconnection of RE. In addition, although RE is an effective means to solve the problems of environmental protection and sustainable development, it is difficult to realize the centralized large-scale development and utilization of RE using the control and dispatching technology of the traditional power grid [4]. In order to coordinate DG and the power grid and to explore the value of DG for the power grid and its users, it is essential to improve the flexibility, controllability, and economy of power system operation to meet user demand for power quality and reliability. Therefore, the energy Internet (EI) is spontaneously indicated to provide the feasibility for the efficient utilization of RE.

The EI is an integrated application of advanced power electronic technology, information technology, and intelligent management technology. To realize the peer-to-peer exchange and sharing network of two-way energy flow, energy nodes composed of DG acquisition devices, distributed energy storage, and various types of loads (power network, natural gas network, and oil networks, etc.) are interconnected [5,6]. In the existing power system, the production, transmission, and consumption of electric energy are operated independently. Consequently, it is insufficient to support the response rate required of DG. In the EI, the functions of energy are integrated, and the main sources of energy are provided by new energy sources [7]. The efficient consumption of RE is not only the consumption of its electric energy, but also its heating and cooling. At present, the accommodation form of RE mainly focuses on electric energy, which limits its capacity of RE to a certain extent. Therefore, the consideration of an integrated energy is necessary for the consumption of RE. This paper discusses how to solve the problem of positive consumption and the comprehensive utilization of new energy through the EI [8].

The development of the power grid has successively experienced three stages: The traditional power, the alternating current (AC) and direct current (DC) hybrid micro-grid, and the EI. The EI is a new form of in-depth integration of energy systems, markets, and Internet technology. It has the advantage of electric energy as an efficient conversion medium for various energy sources with the characteristics of clean priority, multi-energy coordination, and sharing. In order to obtain the optimization of the network distribution in different regions and the allocation of comprehensive energy (cooling, heating, and electricity), a clean energy router (CER) with various energy conversion and routing functions could be a core link to realize energy interconnection, shared production, and distribution.

Similarly to the EI, the aim of the integrated energy system can act as the large-scale development of RE to realize the significant improvement of energy utilization efficiency. Their ultimate purpose is to solve the problems of a sustainable energy supply and environment [9]. As an important part of the integrated energy system, the energy hub model was proposed by the research team of the Swiss Federal Institute of Technology in Zurich [10,11], consisting of the energy transformation and storage of the energy hub and the energy transmission of the energy interconnector. This model can act as a foundation for subsequent research of the integrated energy system, such as the management of the DG system [12] and the operation dispatch of the regional energy system [13]. The concept of energy routers was proposed by a research team at Carolina State University in the U.S., built on the research of the EI based on the construction of new RE and distributed energy storage devices [14]. However, most of the above research mainly considered the coupling relationship between natural gas and the electric power system, lacking consideration of carbon-free clean energy. This paper proposes a clean, safe, and efficient CER based on the energy storage system (ESS), which can improve the comprehensive energy utilization (CEU) rate and can meet various energy demands. In References [15], the authors indicate that the design of an energy router with a single energy conversion function cannot meet the diverse energy needs of users. Compared to the current large-scale ESS, existing mature large-scale ESSs such as pumped hydroelectric storage system (PHS) are limited by the previously mentioned problem and their environmental impact [16]. Moreover, the battery energy storage system (BES) needs to consider the recycling problem cannot meet the requirements of clean energy. Meanwhile, the application of the CER is a lack of research, and the design of the CER based on an advanced adiabatic compressed air energy storage system (AA-CAES) can provide solutions to the above problems. In the information network, the information processor is one of the core technologies of the Internet to conduct the reception, conversion, calculation, storage, and output of a large amount of complex information. The similar functions to the CER mainly include the following three aspects: (1) Considering the source side, it can integrate wind and solar fluctuating power to act as the function of the distributed power source and can convert the input power supply into the stable power output required by users through the CER. (2) Considering the load side, it can provide the flexible conversion of different forms of energy for the requirements of users and can improve the CEU rate. (3) The electric energy on the output side can be divided into AC and DC power distribution to provide a comprehensive supply for users. DC power distribution can be directly connected to DC loads such as DC motors and electric vehicle charging piles, while AC power distribution can be applied to refrigerators and air conditioners that use frequency conversion technology. The overall efficiency of the system reaches 93.6% from the perspective of the CEU rate of the combination of cooling, heating, and electric energy.

This paper illustrates a CER based on AA-CAES, executes the operation mode of the system, and analyzes the efficiency based on the thermodynamic simulation model of AA-CAES, which explores the internal energy conversion mechanism of the system and improves the overall efficiency. The remaining part of this paper is organized as follows. A description of AA-CAES is provided in Section 2. The CER is presented in Section 3. A thermodynamic simulation model of AA-CAES is outlined in Section 4. The energy flow analysis and conversion mechanism of the CER are discussed in Section 5. The simulation and results are presented in Section 6 and the conclusions are provided in Section 7.

## 2. AA-CAES System

AA-CAES is a kind of physical energy with no combustion and zero carbon emissions based on a compressed air energy storage system (CAES) through the recovery of thermally generated compression in the compression process. AA-CAES has the advantages of being clean, efficient, and large-scale. This paper used the seven-stage compression and two-stage expansion of the design for an optimal configuration.

### 2.1. The Working Principle of AA-CAES

Figure 1 demonstrates the decoupling of electric energy into air potential energy and thermal energy through the compressor and heat exchanger to complete the ESS, and then couples the stored air potential energy and thermal energy into electric energy through the turbine to complete energy release [17,18]. The electric energy in the process of the energy storage can use the power from the power grid during off-peak times as well as the wind and solar power stations when the grid connection is limited. The electric energy generated in the process of the energy release can be used for support services, such as grid peak shaving.

Compared to other ESS technologies such as PHS and BES, AA-CAES is not only a clean energy storage (zero carbon emissions and no pollution) that does not damage the ecological environment, but it also has the functions of multi-energy reserves and supplies. Therefore, AA-CAES can be used in the regional thermoelectric integrated energy system. While exerting the synergistic effect of multi-energy complementation, it can enhance the flexibility of the power system and can become an approach to the consumption of wind and solar energy to improve the sustainable high-proportion acceptance capacity of RE.

### 2.2. Compound CAES

Combined with the resource characteristics of the output, AA-CAES can be regarded as a compound CAES. Energy conversion and storage are the basic functions of compound CAES. Therefore, compound CAES can be easily coupled with other systems for the consumption, storage, and comprehensive supply of multiple forms of energy. Figure 2 illustrates that the compound CAES can absorb and store various forms of inferior electric energy through the compressor. It can also absorb and store various forms of thermal energy through the cascade thermal storage system (CTS). The molten salts in CTS are binary salts and a mixture of 60% sodium nitrate and 40% potassium nitrate. Conversely, it can output electric energy with stable voltage and power to provide a support service to the power grid through the process of the expansion. Meanwhile, the exhaust temperature of the turbine can be controlled at 0 °C or even lower to provide an air-conditioned cold air supply for users through the optimization design. Moreover, the CTS can provide users with a thermal energy supply at different temperature levels. It can also drive an absorption chiller to provide indirect cooling for users. The high-pressure air stored in the compound CAES is also an industrial air source that can provide a clean compressed air supply for users through pipelines.

Based on the function of the multi-energy supply shown in Figure 2, compound CAES is the core for building the structure of the CER to provide the energy optimization regulation and comprehensive supply services for the regional micro-energy network. It can couple multiple energy sources by integrating centralized energy (coal power and gas units) and DG systems to meet the diverse energy needs of users through energy conversion between cooling, heating, and electric energy. The function of ESS is mainly reflected in the storage of electricity from the power grid during off-peak times to meet the demand for cooling, electric, and thermal energy of different grades during peak times. The application of compound CAES can improve CEU efficiency while absorbing the thermal waste from geothermal, solar thermal, and industrial production. At the same time, it can be converted into different grades of thermal energy, thereby realizing comprehensive cascade utilization of thermal energy to improve the efficiency of regional CEU.

## 3. The CER

### Coupling Architecture and Working Mode

The main concept of constructing the CER based on AA-CAES is to use the power from the power grid during off-peak times to store the energy in the form of air potential energy and cascade thermal energy through the AA-CAES system, which can be used for peak time power and thermal energy supply. Figure 3 shows the CER coupling composed of AA-CAES and the CTS, which plays a role in the connection route between the energy input and supply.

The main function of AA-CAES is the decoupling of electric energy storage and the peak shaving of the electric energy supply. In the process of energy storage, the compressors decouple the electric energy into potential energy and thermal energy in which the potential energy is stored in the air storage tank (AST) with high-pressure air as the carrier, while the thermal energy is stored in the CTS with thermal storage medium as the carrier. In the input side, the input electric energy mainly consists of the off-peak electric energy of the grid and the decoupled input mode of electric energy. It is also suitable for the power abandoning of wind, solar, and other RE power. During the peak shaving, the CTS supplies high-grade thermal energy coupled with the high-voltage potential in the AST to jointly drive the turbine to output electric energy and to supply the regional energy network (REN). Moreover, the low-temperature air generated in the process of air expansion can provide for industrial and residential users.

The main function of the CTS is to provide a high-grade thermal energy supply for the peak shaving and to provide an uninterrupted industrial steam and hot water supply for the REN. Simultaneously, with the progress of AA-CAES, the other part of the electric energy is directly converted into the high-grade thermal energy through electrical heating and enters the CTS, together with the low-grade thermal energy from AA-CAES. During the peak shaving of AA-CAES, the high-grade thermal energy is used to heat the expander intake air to produce the high-grade electric energy. In the meantime, it still contains the medium-grade thermal energy when the medium-grade thermal energy returns form AA-CAES to the CTS. The CTS uses the medium-grade thermal energy for the continuous supply of steam at 210 °C and the low-grade thermal energy for the continuous supply of hot water at 80 °C.

The energy supply process of the CER is as follows. The energy storage mode is entered during off-peak times, and the compressor and the CTS continuously absorb the electric energy for eight hours during off-peak times. At the same time, the REN directly connects to the local power grid that supplies electricity, while the CTS directly uses the off-peak power to produce steam and hot water for the REN. Once off-peak times end, the power load in the REN is directly supplied by the grid, but the steam and hot water are generated and supplied by the thermal energy stored in the CTS. During peak times, the turbine starts to generate electricity and the REN is disconnected from the power grid. The turbine provides continuous peak shaving power to the REN for four hours during peak times, and then shuts down. Then, the REN is reconnected to the power grid.

## 4. Thermodynamic Simulation Model of AA-CAES

This section proposes a thermodynamic simulation model under the rated operating conditions. Based on the workflow of AA-CAES, the quasi-steady-state thermodynamic AA-CAES simulation is given according to the sequence of the compressor, heat exchanger, thermal storage, AST, and turbine modules.

### 4.1. Compressor Module

The outlet temperature of the *i*-th stage of the compressor is:
(1)$$T_{c,i}^{out}=\frac{1}{\eta_{c,i}}T_{c,i}^{in}\left({\left(\beta_{c,i}\right)}^{\frac{k-1}{k}}+\eta_{c,i}-1\right)$$
where $T_{c,i}^{in}$ is the *i*-th stage of the compressor inlet air temperature, $\beta_{c,i}$ is the *i*-th stage of the compressor pressure ratio, k is the adiabatic index (1.4), and $\eta_{c,i}$ is the *i*-th stage of the compressor isentropic efficiency.

The actual temperature and power of the *i*-th stage of the compressor can be formulated by:
(2)$$p_{c,i}^{out}=p_{c,i}^{in}\beta_{c,i},$$
(3)$$W_{c,i}={\dot{m}}_{c}c_{p}^{a}\left(T_{c,i}^{out}-T_{c,i}^{in}\right),$$
where $p_{c,i}^{in}$ is the inlet air pressure of the *i*-th stage of the compressor, ${\dot{m}}_{c}$ is the compressor mass flow rate, and $c_{p}^{a}$ is the constant pressure-specific heat of the air.

To realize the continuous change of back-pressure in the actual working state of a multi-stage compressor, the operating pressure range of a determined large-capacity AA-CAES with the energy storage is composed of the working back-pressure of the multi-stage compressor system. The compressor power consumption in the case of stable back-pressure can be used in Formula (4), and the compressor power consumption in the case of the continuous rise of back-pressure needs to be integrated over the entire unsteady compression process.
(4)$$W^{\prime }=\int_{p_{1}}^{p_{2}}\frac{1}{\eta_{c}}\frac{kV_{AST}{\dot{m}}_{c}}{k-1}\frac{T_{in}}{T_{cav}}\left[{\left(\frac{p_{out}}{p_{in}}\right)}^{\frac{k-1}{k}}-1\right]dp_{AST},$$
where $p_{AST}$ is the pressure of the AST, $V_{AST}$ is the volume of the AST, and $p_{1}$ and $p_{2}$ are the lower and upper limits of pressure, respectively.

The total power consumption of the compressor is:
(5)$$W_{c}=\sum_{i=1}^{N_{c}}W_{c,i},$$
where $N_{c}$ is the *i*-th stage of the compressor, and $W_{c,i}$ is the actual power of the *i*-th stage of the compressor.

### 4.2. Heat Exchanger Module

During the operation of the system, the heat transfer coefficient of the heat exchanger is easily affected by the heat flow. Therefore, the heat conduction equation should be considered when establishing the various working conditions of the heat exchanger in the designed model [19,20].

#### 4.2.1. Heat Exchanger on the Compression Side

The outlet air temperature of the heat exchanger and the outlet temperature of the heat transfer fluid (HTF) can be calculated, respectively, as:
(6)$$T_{c,HX,i}^{a,out}=T_{c,HX,i}^{a,in}-\frac{\phi_{c,i}^{HX}}{\left(c_{p}^{a}{\dot{m}}_{c,i}^{a}\right)},$$
(7)$$T_{c,HX,i}^{HTF,out}=T_{c,HX,i}^{HTF,in}+\frac{\phi_{c,i}^{HX}}{\left(c_{p}^{HTF}{\dot{m}}_{c,i}^{HTF}\right)},$$
where $T_{c,HX,i}^{a,in}$ is the inlet temperature of the *i*-th stage of the heat exchanger on the compression side, $T_{c,HX,i}^{HTF,in}$ is the inlet temperature of the *i*-th stage of the HTF on the compression side, ${\dot{m}}_{c,i}^{a}$ is the mass flow of the *i*-th stage of the compressor, $c_{p}^{HTF}$ is the constant pressure-specific heat of the HTF, and $\phi_{c,i}^{HX}$ is the actual heat capacity of the HTF on the compression side.
(8)$$\phi_{c,i}^{HX}=\varepsilon_{c,i}\cdot C_{c,i}^{min}\left(t\right)\left(T_{c,HX,i}^{a,in}-T_{c,HX,i}^{HTF,in}\right),$$
where $\varepsilon_{c,i}$ is the heat transfer coefficient of the heat exchanger on the compression side.
(9)$$\varepsilon_{c,i}=\frac{1-exp\left[-NTU_{c,i}\left(1-C_{c,i}^{HX}\right)\right]}{1-C_{c,i}^{HX}exp\left[-NTU_{c,i}\left(1-C_{c,i}^{HX}\right)\right]},$$
where $NTU_{c,i}$ and $C_{c,i}^{HX}$ are the heat transfer unit and the heat capacity ratio of the heat exchanger on the compression side [20], respectively.
(10)$$NTU_{c,i}=\frac{UA}{C_{c,i}^{min}},$$
(11)$$C_{c,i}^{HX}=\frac{C_{c,i}^{min}}{C_{c,i}^{max}},$$
where U and A are the heat exchange coefficient and the heat exchange size, respectively, and $C_{c,i}^{min}$ and $C_{c,i}^{max}$ are the maximum and minimum heat capacities of the heat exchanger on the compression side, respectively.
(12)$$C_{c,i}^{min}={\left({\dot{m}}_{c}c_{p}^{a},{\dot{m}}_{c,i}^{HTF}c_{p}^{HTF}\right)}_{min},$$
(13)$$C_{c,i}^{max}\left(t\right)={\left({\dot{m}}_{c}\left(t\right)c_{p}^{a},{\dot{m}}_{c,i}^{HTF}\left(t\right)c_{p}^{HTF}\right)}_{max},$$
where ${\dot{m}}_{c,i}^{HTF}$ is the mass flow of the *i*-th stage of the HTF and can be defined as:
(14)$${\dot{m}}_{c,i}^{HTF}=\frac{{\dot{m}}_{c}c_{p}^{a}\left(T_{c,HX,i}^{a,in}-T_{c,HX,i}^{a,out}\right)}{c_{p}^{HTF}\left(T_{c,HX,i}^{HTF,out}-T_{c,HX,i}^{HTF,in}\right)}.$$


However, the heat exchanger operates under partial load when the mass flow of high-temperature air through the heat exchanger is considered in a practical application [21,22]. Meanwhile, most of the current models assume that the efficiency of the heat exchanger is constant without pressure loss. Pressure loss directly affects the overall efficiency of the system, as the heat exchanger is one of the core components of AA-CAES. The pressure retention coefficient of the heat exchanger on the compression side can be expressed as:
(15)$$\eta_{c,HX,i}^{p}=1-\frac{0.0083\varepsilon_{c,i}}{1-\varepsilon_{c,i}}.$$


The outlet pressure of the heat exchanger on the compression side is:
(16)$$p_{c,HX,i}^{out}=\eta_{c,HX,i}^{p}p_{c,HX,i}^{in}.$$
where $p_{c,HX,i}^{in}$ is the inlet air pressure of the *i*-th stage of the heat exchanger on the compression side.

#### 4.2.2. Heat Exchanger on the Expansion Side

Similar to the heat exchanger on the compression side, the outlet temperature of the heat exchanger and the HTF on the expansion side can be formulated, respectively, as:
(17)$$T_{e,HX,i}^{a,out}=T_{e,HX,i}^{a,in}+\frac{\phi_{e,i}^{HX}}{\left(c_{p}^{a}{\dot{m}}_{e}^{a}\right)},$$
(18)$$T_{e,HX,i}^{HTF,out}=T_{e,HX,i}^{HTF,in}-\frac{\phi_{e,i}^{HX}}{{\left(c_{p}^{HTF}{\dot{m}}_{e,i}^{HF}\right)}^{\prime }}$$
where $T_{e,HX,i}^{a,in}$ is the inlet temperature of the *i*-th stage of the heat exchanger on the expansion side, $T_{e,HX,i}^{HTF,in}$ is the inlet temperature of the *i*-th stage of the HTF on the expansion side, ${\dot{m}}_{e}^{a}$ is the mass flow of the *i*-th stage of the turbine, ${\dot{m}}_{e,i}^{HTF}$ is the mass flow of the HTF on the expansion side, and $\phi_{e,i}^{HX}$ is the actual heat capacity of the HTF on the expansion side.
(19)$$\phi_{e,i}^{HX}=\varepsilon_{e,i}\cdot C_{e,i}^{min}\left(t\right)\left(T_{e,HX,i}^{a,in}-T_{e,HX,i}^{HTF,in}\right),$$
where $\varepsilon_{e,i}$ is the heat transfer coefficient of the heat exchanger on the expansion side.
(20)$$\varepsilon_{e,i}=\frac{1-exp\left[-NTU_{e,i}\left(1-C_{e,i}^{HX}\right)\right]}{1-C_{e,i}^{HX}exp\left[-NTU_{e,i}\left(1-C_{e,i}^{HX}\right)\right]},$$
where $NTU_{e,i}$ and $C_{e,i}^{HX}$ are the heat transfer unit and the heat capacity ratio of the heat exchanger on the expansion side [23], respectively.
(21)$$NTU_{e,i}=\frac{UA}{C_{e,i}^{min}},$$
(22)$$C_{e,i}^{HX}=\frac{C_{e,i}^{min}}{C_{e,i}^{max}},$$
where U and A are the heat exchange coefficient and the heat exchange size, respectively, and $C_{e,i}^{min}$ and $C_{e,i}^{max}$ are the maximum and minimum heat capacities of the heat exchanger on the expansion side, respectively.
(23)$$C_{e,i}^{min}={\left({\dot{m}}_{e}c_{p}^{a},{\dot{m}}_{e,i}^{HTF}c_{p}^{HTF}\right)}_{min},$$
(24)$$C_{e,i}^{max}\left(t\right)={\left({\dot{m}}_{e}c_{p}^{a},{\dot{m}}_{e,i}^{HTF}c_{p}^{HTF}\right)}_{max},$$
where ${\dot{m}}_{e,i}^{HTF}$ is the mass flow of the *i*-th stage of HTF and can be defined as:
(25)$${\dot{m}}_{e,i}^{HTF}=\frac{{\dot{m}}_{e}c_{p}^{a}\left(T_{e,HX,i}^{a,in}-T_{e,HX,i}^{a,out}\right)}{c_{p}^{HTF}\left(T_{e,HX,i}^{HTF,out}-T_{e,HX,i}^{HTF,in}\right)},$$


Considering the pressure loss characteristics of the heat exchanger, the pressure retention coefficient of the heat exchanger on the expansion side can be expressed as:
(26)$$\eta_{e,HX,i}^{p}=1-\frac{0.0083\varepsilon_{e,i}}{1-\varepsilon_{e,i}}.$$


The outlet pressure of the heat exchanger on the compression side is:
(27)$$p_{e,HX,i}^{out}=\eta_{e,HX,i}^{p}p_{e,HX,i}^{in}.$$
where $p_{e,HX,i}^{in}$ is the inlet air pressure of the *i*-th stage of the heat exchanger on the expansion side.

According to the law of conversation of mass, the mass flow rate and temperature of the HTF can be calculated, respectively, as:
(28)$${\dot{m}}^{HTF}=\sum_{i=1}^{N}{\dot{m}}_{i}{}^{HTF},$$
(29)$$T_{HX}^{Merge}=\frac{\sum_{i=1}^{N}{\dot{m}}_{HX,i}^{HTF}T_{HX,i}^{HTF,out}}{\sum_{i=1}^{N}{\dot{m}}_{HX,i}^{HTF}}-T_{HX,i}^{HTF,in}.$$


### 4.3. Thermal Storage Module

Due to the difference in compressor grades and high-pressure air temperature at the compressor outlet, AA-CAES heat storage devices can be divided into high-temperature [24], medium-temperature [25], and low-temperature [26] thermal storage tanks. Due to the thermal storage medium of the thermal storage device being different, thermal storage technology can be divided into four types including fouble-tank liquid thermal storage [27], concrete thermal storage [28], molten salt thermal storage [29], and phase-change material thermal storage [30]. The temperature of the thermal storage medium can be described as:
(30)$$\left(\rho_{TES}V_{TES}c_{p}^{TES}\right)\frac{dT_{TES}}{dt}={\dot{m}}_{c}^{HTF}c_{p}^{HTF}\left(T_{c}^{TES,in}-T_{TES}\right)-{\dot{m}}_{e}^{HTF}c_{p}^{HTF}\left(T_{TES}-T_{e}^{TES,in}\right)-Q,$$
where $\rho_{TES}$ is the heat storage medium density, $V_{TES}$ is the heat storage medium volume, $c_{p}^{TES}$ is the constant pressure-specific heat of the thermal energy storage, $T_{c}^{TES,in}$ is the inlet temperature of the heat storage medium in the compression process, $T_{e}^{TES,in}$ is the inlet temperature of the heat storage medium in the expansion process, and Q is the heat transfer with surrounding air, which can be defined as:
(31)$$Q=U_{TES}A_{TES}\left(T_{TES}-T_{env}\right),$$
where $U_{TES}$ and $A_{TES}$ are the heat transfer coefficients between the heat storage tank and the environment and the external surface area of the heat storage tank ($U_{TES}$ is 0 when the heat transfer loss is not considered), and $T_{env}$ is the temperature of the environment.

### 4.4. AST Module

In this system, the AST is the main equipment of the energy storage. The current AST approaches mainly include constant pressure AST [30] and constant capacity AST [31]. To accurately simulate the instantaneous characteristics of the temperature and pressure in the AST, a constant volume AST with less geographical dependence was selected in this paper. The conservation of air mass and the generalized air state equation can be defined, respectively, as:
(32)$$V\frac{d\rho }{dt}=a,$$
(33)p=ZρRT.
where Z is the compressibility factor of air, ρ is the density of air, R is the gas constant value, and T is the air temperature of the AST.

During the operating period of AA-CAES, the air mass flow can be defined as the constant. Therefore, the ordinary differential equation of air temperature in the AST is:
(34)$$T=\{\begin{array}{c} \left(T_{0}+\frac{m_{c}C_{p}T_{i}+h_{c}A_{c}T_{RW}}{m_{c}\left(R-C_{p}\right)-h_{c}A_{c}}\right)e^{\frac{m_{c}\left(R-C_{p}\right)-h_{c}A_{c}}{V\rho_{av}c_{v}}\left(t-t_{0}\right)}-\frac{m_{c}C_{p}T_{i}+h_{c}A_{c}T_{RW}}{m_{c}\left(R-C_{p}\right)-h_{c}A_{c}},a=1\\ \left(T_{0}-T_{RW}\right)e^{\frac{-h_{c}A_{c}}{V\rho a_{v}c_{v}}\left(t-t_{0}\right)}+T_{RW},\;a=0\\ \left(T_{0}+\frac{h_{c}A_{c}T_{RW}}{m_{c}R-h_{c}A_{c}}\right)e^{\frac{m_{c}R-h_{c}A_{c}}{V\rho a_{v}c_{v}}\left(t-t_{0}\right)}-\frac{h_{c}A_{c}T_{RW}}{m_{c}R-h_{c}A_{c}},\;a=-1 \end{array}$$
where $h_{c}$ is the average heat transfer coefficient for the AST, $A_{c}$ is the surface area for AST, $T_{RW}$ is the surface temperature of the AST, $\rho_{av}$ is the average density of the air under the whole operation progress, $c_{v}$ is the specific heat capacity of the air, $T_{R}$ is the internal temperature of the AST, $t_{0}$ is the initial time, $T_{i}$ is the inlet temperature of the *i*-th stage of the AST, $T_{0}$ is the initial air temperature, with a=1, a=0, and a=−1 representing the charging period, idle period, and discharging, respectively.

### 4.5. Turbine Module

The actual output temperature of the *i*-th stage of the turbine is:
(35)$$T_{e,i}^{out}=T_{e,i}^{in}\left(1-\eta_{e,i}+\eta_{e,i}{\left(\beta_{e,i}\right)}^{\frac{1-k}{k}}\right),$$
where $T_{e,i}^{in}$ is the *i*-th stage of the turbine inlet air temperature, $\beta_{e,i}$ is the *i*-th stage of the turbine pressure ratio, k is the adiabatic index (1.4), and $\eta_{e,i}$ is the *i*-th stage of the turbine isentropic efficiency.

The actual power and output pressure of the *i*-th stage of the turbine can be formulated, respectively, as:
(36)$$W_{e,i}={\dot{m}}_{e}c_{p}^{a}\left(T_{e,i}^{in}-T_{e,i}^{out}\right),$$
(37)$$p_{e,i}^{out}=\frac{p_{e,i}^{in}}{\beta_{e,i}},$$
where $p_{e,i}^{in}$ is inlet air pressure of the *i*-th turbine, $p_{e,i}^{out}$ is the outlet air pressure of the *i*-th turbine, and ${\dot{m}}_{e}$ is the turbine mass flow rate.

The total power of the turbine is:
(38)$$W_{e}=\sum_{i=1}^{N_{e}}W_{e,i},$$
where $N_{e}$ is the *i*-th stage of the turbine, and $W_{e,i}$ is the actual power of the *i*-th stage of the turbine.

## 5. Energy Flow Analysis and Conversion Mechanism

The CER can realize the storage and conversion of electric energy to achieve the comprehensive cooperative supply of multiple energies. In the process of energy storage, first, the CER uses AA-CAES technology to convert electric energy into pressure potential energy and compressed thermal energy of high-pressure air to obtain the decoupling storage of the pressure potential energy and compressed thermal energy of electric energy. Second, the CER directly converts electric energy into high-grade thermal energy through the CTS and realizes the cascade storage of thermal energy in the multi-temperature region.

In the process of energy release, according to different load requirements, the CER can provide different forms of energy such as electric, thermal (industrial steam and hot water), and cooling energy. First, the CER can couple the pressure potential energy of high-pressure air with the high-grade thermal energy in the CTS to achieve high-efficiency energy release and power generation. Second, the medium-grade thermal energy in the CTS is used to produce steam to meet industrial demand. Third, the low-grade thermal energy in the CTS is used to produce hot water to meet the needs of daily life. Through the cascade utilization of thermal energy, the CEU and the system efficiency can be improved.

### 5.1. Heating Energy Produced

The storage of high-pressure air is achieved by using off-peak power, abandoned wind, and solar power to drive the compressor. The compressor adopts seven-stage series compression. During the compression process, the pressure in the AST changes, which is followed by the compression time. Therefore, the exhaust pressure of the final stage changes, followed by the compression time, and results in the unstable condition of the compressor. With the increase in pressure, the power consumption of the compressor increases gradually. Through Formula (5), we can obtain the total energy consumption of the compressor when the CTS works as well as the compressor.

The thermal storage process is mainly composed of two parts: One is to convert electric energy into thermal energy by electric heating of molten salt, and the other is to use the high-temperature thermal energy in the molten salt to increase the intake temperature of the expansion generator through a heat exchanger in order to improve the efficiency of power generation. When using molten salt to store thermal energy, there is a 2% loss and the minimum temperature difference of the molten salt heat exchanger is 30 °C. The electric energy consumed by the electric heating of molten salt is:
(39)$$E_{TES}=t_{s}Q_{m,s}c_{p}^{c}\left(T_{TES,H}-T_{TES,L}\right),$$
where $t_{s}$ is the electric heating time, $Q_{m,s}$ is the flow rate of the molten salt during electric heating, $c_{p}^{c}$ is the specific heat capacity of molten salt, and $T_{TES,H}$ and $T_{TES,L}$ are the outlet and inlet temperatures of electric heater, respectively.

The high-pressure air entering each stage of the turbine is heated through the molten salt. The thermal energy absorbed by the high-pressure air can be calculated by:
(40)$$Q_{e,i}=t_{e}Q_{m,e}c_{p}^{a}\left(T_{e,i}-T_{e,i0}\right),$$
where $T_{e,i0}$ is the air temperature before entering the heater, $t_{e}$ is the heating supply time, and $Q_{m,e}$ is the flow rate of the molten salt during the heating supply. Since the system adopts a two-stage series expansion generator, the total thermal utilized by the high-pressure air during the generation process is:
(41)$$Q_{e}=\sum_{i=1}^{N_{e}}Q_{e,i}.$$


### 5.2. Cooling Energy Produced

In Equation (38), the total power by the turbine can be obtained. When the turbine works, cooling energy with the cooler function is generated. Therefore, the cooling amount in the process is:
(42)$$Q_{c,i}=t_{e}Q_{m,e}c_{p,a}\left(T_{e,i}-T_{e,i0}\right).$$


### 5.3. Overall Energy Storage Efficiency

The ratio between the energy generated in the energy release process and the energy consumed in the energy storage process of the system is the energy storage efficiency of the system, and its expression is:
(43)$$\eta =\frac{W_{e}+Q_{c,i}+Q_{e,i}}{W_{c}+E_{TES}}.$$


AA-CAES is an efficient and clean large-scale energy storage technology. The CER based on AA-CAES designed the storage and mutual conversion mechanism of clean energy such as wind, light, electricity, and thermal energy. Therefore, the CER has the ability to output the cooling, heating, and electric energy to residents, businesses, and industries. Based on the calculations, the CEU efficiency of the whole system exceeds 90%. The specific energy flow distribution will be verified in subsequent simulation research.

## 6. Simulation and Results

To verify the model, the modeling and operation based on the AA-CAES system mainly assumed that air is an ideal gas and satisfies the ideal gas state equation, and that the specific heat capacity of air for the heat transfer medium and the heat storage medium are constant. The main parameters of the system are shown in Table 1. Meanwhile, to further clarify the performance of the proposed system, a comparison of AA-CAES to a combined cooling, heating, and power (CCHP) system using the internal combustion engines and gas turbine in Reference [32] is demonstrated in Table 2. In the comparison case, the round-trip efficiency for each system is 96.5% and 87%, respectively. As integrated energy systems for cooling, heating, and power supply, they have the characteristics of providing diverse energy sources and high round trip efficiency. The CCHP system requires the combustion of natural gas and the consumption of electric energy, which brings a large amount of carbon emissions. The CER system based on the AA-CAES system proposed in this paper only requires the consumption of electric energy during operation, which have the advantage of clean energy without environmental pollution.

### 6.1. Charging Process

In the compression process, the ambient temperature of the system was 293 K, and the initial pressure in the AST was 2.82 MPa. Under the condition of rated parameters, the compression process was compared to the adiabatic and heat exchange of the AST. In the operation process, the outlet pressure of all of the levels of the compressors changed over time for the two modes of AST adiabatic and AST heat transfer, as shown in Figure 4a,b. This demonstrates that the air storage process was simulated for 8 h. The first five-stage compressor units worked stably throughout the whole process, while the sixth compressor reached the stable state under the condition of variable back-pressure, and the seventh compressor worked with variable back-pressure throughout the whole process. Multi-stage compression was able to reduce the energy consumption and improve the efficiency of the system. When considering the heat exchange of the AST, the sixth stage compressor reached a stable pressure within 3.4 h compared to 2.5 h in the adiabatic condition of the AST. The sixth stage reached stable pressure faster under adiabatic conditions than under heat exchange conditions because the heat and pressure loss under adiabatic conditions were not considered. The adiabatic conditions of the AST were closer to the actual situation when we need to consider the comparison with the actual data of the system.

Figure 5a,b illustrate the power consumption of each stage for the two modes of AST adiabatic and AST heat exchange during the compression process. The changing characteristics of the power consumption mainly reflect in the changing back-pressure time of the sixth and seventh stages, which is the same as the pressure change curve in Figure 4a,b.

### 6.2. Discharging Process

In the process of expansion, the change in outlet pressure of all levels of the turbines with expansion time is shown in Figure 6a. Since the inlet pressure of the turbine expansion generator was adjusted from the throttle valve to the given pressure value, the adiabatic and heat exchange conditions of the AST did not influence the outlet pressure on the expansion side. The outlet pressure of the first and second-stage expander was 0.8614 MPa and 0.1025 MPa, respectively, which meet the design requirements of the system. The outlet pressure after the first-stage expansion generator worked was directly supplied to the inlet pressure of the second-stage expansion generator, thus, driving the second-stage expansion generator to generate electricity.

Figure 6b illustrates the temperature change in the medium-temperature molten salt thermal storage device that received heat recovery from the expansion generator. During the 4-h generation process, the temperature change of the adiabatic heat exchange and external heat exchange of the AST are presented, respectively. It is essential to consider the heat exchange of the medium-temperature thermal storage unit due to the accuracy of the model.

### 6.3. The Whole System Operation Process

To verify the integrity of the system, the temperature and pressure changes of the compressed AST were simulated by using 24 h as the period shown in Figure 7. The pressure increased to 8.92 MPa in 8 h with a change in time. After 2 h of idle time, the pressure remained almost stable. After 4 h of power generation, the pressure of the AST decreased to 2.82 MPa and remained up until 10 h of idle time. After the change of temperature reached 315.8 K in 8 h, it then became stable. After 2 h, the temperature in idle time gradually dropped to 313.5 K. The temperature dropped again to 309.8 K after generating power. Finally, the temperature gradually returned to 313.5 K in the last 2 h.

The overall efficiency of the system can be analyzed in two situations. One is the electric to electric efficiency and the second is the CEU efficiency, which combined cooling, heating, and power. The system used high-pressure air as the working medium to store electric energy for 8 h and then 2 h of idle time, and used heated high-pressure air by the heat exchanger to drive the generator to generate electricity for 4 h when required by the users (shown in Figure 8). The efficiency of the power exchange was obtained by the ratio between the power consumption and the power generation, and was shown to be 56.5%.

The heat exchange of the compression side of the heat exchanger is shown in Figure 9. When the system stored air for 8 h, it achieved 80 °C hot water through the heat exchange between water and the heat exchanger. The remaining 16 h supply of hot water was used by electric heating. At the same time, 16 h of high-temperature industrial steam could be supplied through electric heating to support regional industrial demand. With regard to the operation of the power generator, the cooling energy for driving the generator could be used to supply the cold demand (air conditioners, refrigerators, etc.) of the residents, industry, and commerce in the region. The cooling capacity can be obtained based on the exhaust temperature (10 °C) and operation time of the turbine. Therefore, the overall efficiency of the combination of the cooling, heating, and electric energy reached 93.6%. This not only ensures the clean and safe supply of energy, but it also reflects the efficient and economical performance of the system. The promotion of using the CER can meet the demand for users, realize the optimal allocation of a variety of energy resources, and improve the energy efficiency of users.

## 7. Conclusions

In order to solve the consumption and storage of abandoned wind power, abandoned solar power, and electricity during off-peak times, a CER based on AA-CAES was proposed and analyzed in this paper. The characteristics of the CER include the combination of cooling, heating, and electric energy. It can transform the stored electric energy into various types of clean energy such as the high-grade electric energy, cold energy, and different cascade thermal energy. Therefore, the effectiveness of the simulation and the verification of the system were indicated. Some main conclusions of the proposed CER are as follows:
The results show the outlet pressure change of the compressor under the two conditions of the adiabatic and heat exchange of the AST. When considering the heat exchange of the AST, the sixth stage of the compressor reached a stable pressure in 3.4 h compared with 2.5 h under the adiabatic conditions of the AST. The adiabatic conditions of the AST were closer to the actual situation when we need to consider the comparison with the actual data of the system.The results of the discharging process demonstrated that the adiabatic and heat exchange of the AST did not influence the outlet pressure of the turbine, and, thus, meet the requirements of the design. Meanwhile, it is essential to consider the heat exchange of the medium-temperature thermal storage unit due to the accuracy of the model.Finally, the efficiency of the system was analyzed based on two aspects: First, the efficiency of the power exchange was obtained by the ratio between the power consumption and the power generation as 56.5%. Second, the overall efficiency of the system reached 93.6% from the perspective of the CEU rate of the combination of cooling, heating, and electric energy.


## Figures and Tables

**Figure 1 entropy-22-01440-f001:**
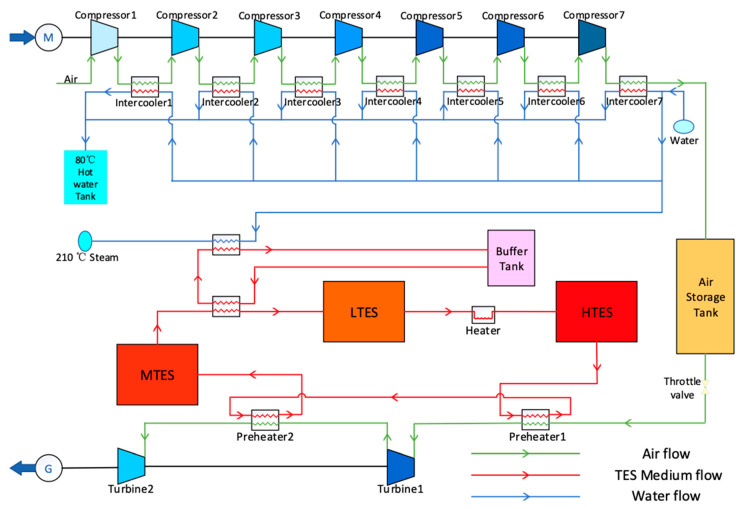
Schematic diagram of advanced adiabatic compressed air energy storage system (AA-CAES).

**Figure 2 entropy-22-01440-f002:**
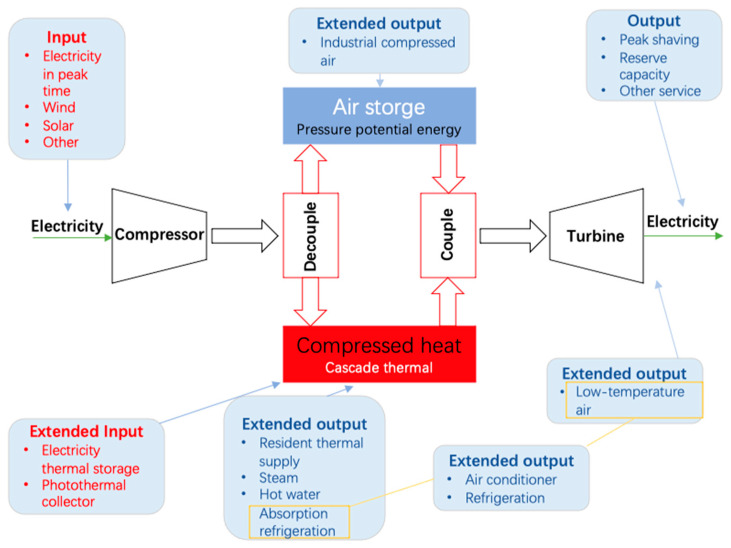
Schematic diagram of the extended characteristics of compound compressed air energy storage system (CAES).

**Figure 3 entropy-22-01440-f003:**
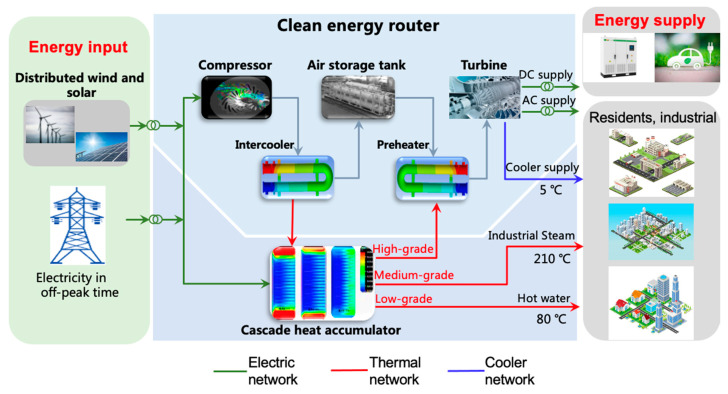
Schematic diagram of the clean energy router (CER).

**Figure 4 entropy-22-01440-f004:**
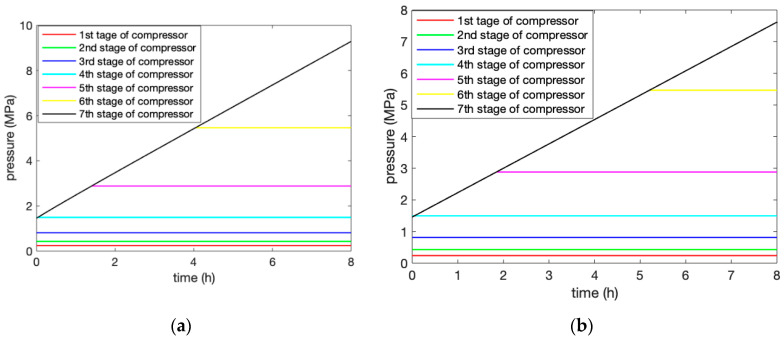
Changes in the outlet pressure of the compressors at all levels during the compression under (**a**) adiabatic and (**b**) heat exchange conditions.

**Figure 5 entropy-22-01440-f005:**
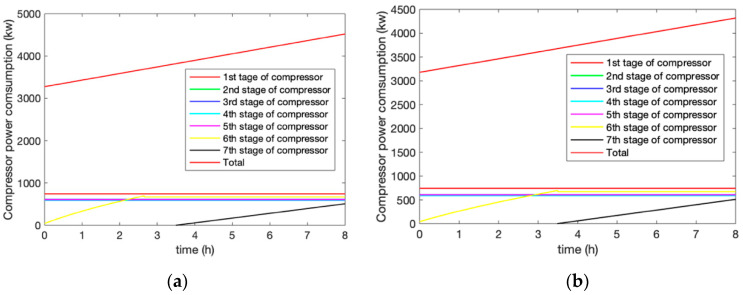
Power consumption of the compressor for each stage of the compression process under (**a**) adiabatic and (**b**) heat exchange conditions.

**Figure 6 entropy-22-01440-f006:**
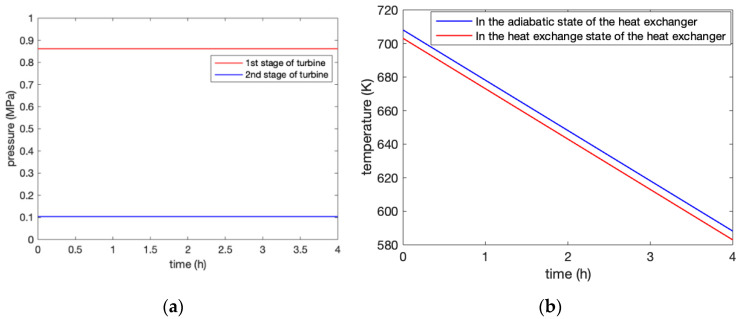
(**a**) Changes in the outlet pressure of the turbines at all levels during the expansion process. (**b**) Temperature variation of the medium-temperature molten salt thermal storage unit.

**Figure 7 entropy-22-01440-f007:**
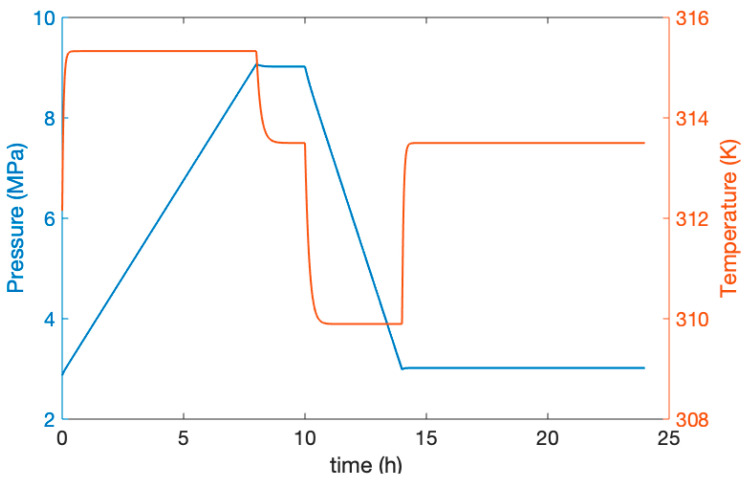
Temperature and pressure variation of the AST during one round-trip.

**Figure 8 entropy-22-01440-f008:**
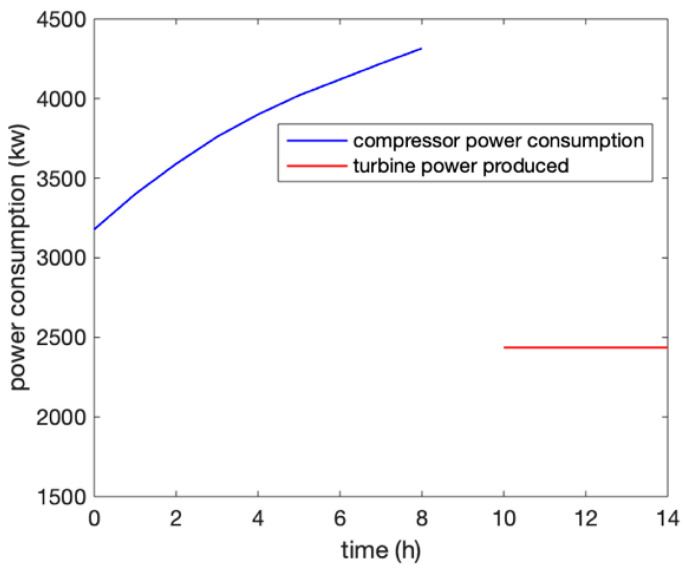
Power consumption and generation in the compression and expansion processes.

**Figure 9 entropy-22-01440-f009:**
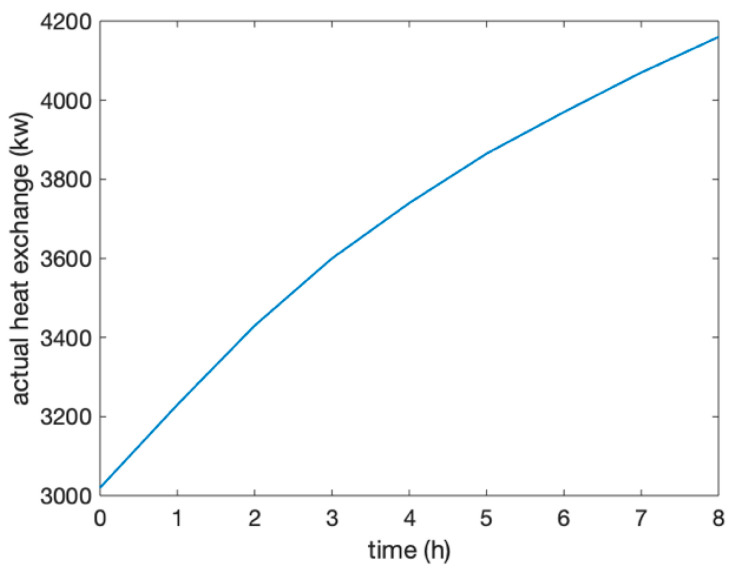
The amount of heat exchange of the heat exchanger in the compression process.

**Table 1 entropy-22-01440-t001:** The rated parameters of the advanced adiabatic compressed air energy storage system (AA-CAES) system.

Parameters	Value	Unit
Compressor isentropic efficiency	0.84	/
Turbine isentropic efficiency	0.9	/
Air mass flow of compressor	7.39	kg/s
Air mass flow of turbine	14.46	kg/s
Air mass flow of thermal storage medium	2.18	kg/s
Specific heat capacity of air at constant pressure	1005	J/kg·k
The specific heat capacity of water	4200	J/kg·k
Air constant	287	J/kg·k
Heat transfer coefficient between air and the outside of the tank	30	W/(m^2^·k)
Ambient temperature	293	K
Ambient pressure	0.1015	MPa
Compression period	8	h
Expansion period	4	h

**Table 2 entropy-22-01440-t002:** The comparison of performance of the AA-CAES to the combined cooling, heating, and power (CCHP) system.

Parameters	AA-CAES	CCHP
Round trip efficiency, %	93.6	87%
Electric to electric efficiency, %	56.5	/
Exergy efficiency, %	/	80%

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
