# Peer review of "Technological Research of a Clean Energy Router Based on Advanced Adiabatic Compressed Air Energy Storage System"

_entropy, 2020, doi:10.3390/e22121440_

Round 1
Reviewer 1 Report
The paper describes a system for the energy storage based on compressed air. The idea behind is potentially good, but I find the paper not well written: many symbols in the equations are not described, the figures are not clear ...
It is difficult for me to follow the paper flow.
Reviewer 2 Report
Authors used this publication as a “citing club”. Self-referencing is a mispractice that should be eradicated from Science otherwise it will discredit journal reputation.
Reviewer 3 Report
The paper 'The Technology Research of Clean Energy Router Based on Advanced Adiabatic Compressed Air Energy Storage' presents an energy storage configuration based on compressed air to take advantage of the electricity produced by renewable sources in an energy grid. The reviewer has the following comments:
a) The topic is an interesting one from the point of grid energy management.
b) What is the pressure ratio across the compressor stages? Was it optimized? How much energy is required to run the compressors? It is recommended to include some figures regarding the energy consumption and characteristics of the compressors.
c) The heat storage is to be charged by electric heater? How much energy is used during this process? If molten salt is used, which salt is it? to what temperature is it heated? How much energy is used in the electric heating process?
d) What about the efficiency of the system? heater temperature and pressure ratio should affect the final efficiency value.
e) Figures must be checked and format to include units on labels.
f) English must be revised.
g) There is a significant quantity of acronyms. it is recommended to reduce the quantity for clarity.
Reviewer 4 Report
The work is very interesting focusing in a very important problem of energy storage generated by inttermittent renewable energies. The idea and analysis of energy storage in pressurized air using the CER and regenerating heat, cooling and electricity is very interesting. Also it would be interesting to mention if there are any demonstration or commercial applications of the concept. Some comments also should be added regarding the comparison of this electricity storage system with others including storage in Hydrogen and in hydro pump storage systems.
Round 2
Reviewer 1 Report
The authors improved the quality of the paper. However, the figures are still not very clear (too small characters).
Reviewer 2 Report
Authors made cosmetic changes to their manuscript, but there are several serious flaws from their work:
- The novelty of the paper is not clear. Authors should highlight what's their science contribution in this publication and compare their results to the existing literature
- Validation of simulation results cannot be checked as solving methodology is not appropiately described. Simulation results should be compared to the existing literature and improvements introduced by authors' contribution should be highlighted
- The paper is full of mistakes regarding units acronyms
- Solving methodology might be wrong as design conditions are imposed (Table 1) for time-dependant resolution (Figure 4-9). Time-dependant solving scheme and its validation is not included
- Authors still abuse of self-citation (20%)
Reviewer 3 Report
Thanks for your response.
Still, I find concerns about the heat storage system. More clearness about which salt is being simulated and the detailed description of the heat storage system should be given.
Also the efficiency of the system is told to be 93.6% when cooling and heating is contemplated, but there is nothing in the simulation and results section that can be used to validate that fact.
